# Downregulation of GABAARα1 Aggravates Comorbidity of Epilepsy and Migraine via the TLR4 Signaling Pathway

**DOI:** 10.3390/brainsci12111436

**Published:** 2022-10-25

**Authors:** Yao Lin, Man Ding, Qiaoyu Gong, Zheman Xiao

**Affiliations:** 1Department of Neurology, Renming Hospital of Wuhan University, Wuhan 430060, China; 2Central Laboratory, Renming Hospital of Wuhan University, Wuhan 430060, China

**Keywords:** animal behavior, comorbidity, epilepsy, GABAARα1, migraine, TLR4

## Abstract

Epilepsy and migraine are among the most prevalent neurological disorders. By being comorbid, the presence of one disorder increases the likelihood of the other. Although several similar clinical features of epilepsy and migraine have been observed as early as the 19th century, only in recent years have researchers engaged in finding a common pathogenic mechanism between them. In this study, the epilepsy–migraine comorbidity rat model was generated, and the pathophysiological basis of epilepsy–migraine comorbidity was examined. Male rats were divided into four groups: control, migraine, epilepsy, epilepsy–migraine comorbidity. After establishing the models, the amount of scratching and the pain threshold of the rats were observed. Western blot and immunofluorescence staining were used to detect the protein expression levels of TLR4 and GABAARα1 in the temporal cortex, hippocampus, trigeminal ganglion, and medullary dorsal horn. Subsequently, co-immunoprecipitation of GABAARα1 and TLR4 was performed. Then, the rats were divided into three groups: comorbidity, comorbidity + TAK-242, and comorbidity + muscimol. After drug intervention, the seizure latency, seizure level, amount of scratching, and pain threshold were observed. Western blot was used to detect the protein expression levels of TLR4 and GABAARα1 in the temporal cortex, hippocampus, trigeminal ganglion, and medullary dorsal horn. Our results demonstrate that the seizure attacks in comorbidity and epilepsy groups performed severely, and the comorbidity and migraine groups displayed a remarkable increase in the amount of head-scratching and a noticeable decrease in the facial mechanical withdrawal threshold. Further analysis revealed considerably increased Toll-like receptor 4 (TLR4), associated with reduced γ-aminobutyric acid type A receptor α1 (GABAARα1) and microglia enhanced in the epilepsy–migraine comorbidity rat. Additionally, co-immunoprecipitation proved GABAARα1 binding TLR4. Following muscimol to activate GABAARα1, seizure attacks and migraine-like behavior were rescued. GABAARα1 level increment was accompanied by the decline of TLR4, while TAK-242, the inhibitor of TLR4, only decreased TLR4 without affecting GABAARα1 expression. It also ameliorated the migraine-like behavior with no impact on seizure activity. We propose that GABAARα1 binding and negatively regulating TLR4 contribute to epilepsy–migraine comorbidity; TLR4 is a critical intermediate link in epilepsy–migraine comorbidity; immune-induced neuroinflammation in microglia may be involved in migraine and epilepsy–migraine comorbidity.

## 1. Introduction

Epilepsy and migraine are both common disorders. Epidemiological studies have confirmed a lifetime prevalence of epilepsy of 4% and an adult prevalence of nearly 1% [1]. About 12% of people have migraines, and 2% of them have chronic migraines [2,3]. According to the Centers for Disease Control, the presence of active epilepsy increased the prevalence of severe headaches or migraines to 35.5% [4]. Adults with epilepsy are 2.4 times more likely to have migraines than their non-epileptic relatives, according to a large epidemiological study [5]. Patients with longer epilepsy duration had a greater risk of occurrence of seizure-related headaches. Both epilepsy and migraine are characterized by sudden, paroxysmal changes in mood and behavior, including changes in consciousness, accompanied by changes in visual, motor, sensory, or language functions. At the time of onset, both are clinically characterized by recurrent episodes, usually occurring before prodrome and/or aura [6]. Epilepsy–migraine comorbidities are classified into epileptic migraine, pre-seizure migraine, post-seizure migraine, and inter-epileptic migraine. Among them, migraine after a seizure is the most common [7,8]. Compared with ordinary patients with epilepsy, the epilepsy symptoms of patients with epilepsy–migraine comorbidity are more difficult to alleviate, and the quality of life will be further reduced [9]. Although several similar clinical features of epilepsy and migraine have been observed as early as the 19th century, only in recent years have researchers really engaged in finding a common pathogenic mechanism between them.

It has been confirmed that epilepsy mainly causes the overexcitation of brain neurons, which is mainly caused by the excitation–inhibition imbalance of neurons mediated by glutamate and γ-aminobutyric acid (GABA) [10,11]. The GABA type A receptors (GABAARs) are the most prevalent GABA receptor (GABAR) in the mammalian brain. Known to α1–3, β2–3 and γ2 subunit compositions play important roles in the many subunits of GABAARs [12,13]. Loss-of-function mutations in GABAAR genes are an essential mechanism in the pathophysiology of familial idiopathic generalized epilepsy. The α1 subunit D219N mutation was found in patients with idiopathic systemic epilepsy [14]. In addition, Maria-Eliza R. Aguila et al. found that higher pain and central sensitization scores were associated with increased brain GABA levels in individuals with migraine [15]. However, GABAARα1 subunit’s direct role in regulating epilepsy–migraine comorbidity and its downstream molecular pathways are unknown. 

Toll-like receptors (TLRs) are a family of pattern-recognition receptors that play an important role in initiating innate and adaptive immune pathways in the body. The receptors are mainly found in inflammatory cells, axonal glial cells, Schwann cells, fibroblasts, dorsal root ganglion, and dorsal horn neurons [16]. Although TLR4 has been shown to play an important role in immune function and inflammatory response, the role of TLR4 antagonists in targeted therapy or prevention of chronic neuropathic pain is still being investigated [17,18].

“Inflammatory soup”(IS) is a mixture of prostaglandin E2 (0.1 mmol/L, Sigma, St. Louis, MO, USA), serotonin (1 mmo/L, Soledad Takara Bio Co., Ltd., Hangzhou, China), bradykinin (1 mmo/L, Soledad Takara Bio Co., Ltd., Hangzhou, China), and histamine (1 mmo/L, Soledad Takara Bio Co., Ltd., Hangzhou, China), which is commonly used in migraine-related studies to detect the response of pain receptors (including the meninges) to immune derivatives released at inflammatory sites [19]. Injection of IS into animal meninges can enhance skin mechanosensation, the sensitivity of the pain receptor to mechanical stimulation, and the sensitization of neurons in the trigeminal caudate nucleus [4]. Therefore, the migraine model of this study was established with IS.

In the present study, we established the epilepsy–migraine comorbidity rat model and investigated that the occurrence of epilepsy–migraine comorbidity is associated with a GABAARα1-TLR4 pathway in different brain sites. For the first time, we investigated the downstream role of the innate immune receptor TLR4 in GABAARα1 and combined with the epilepsy–migraine comorbidity, providing a meaningful research direction.

## 2. Materials and Methods

### 2.1. Animals and Experimental Grouping

Male Sprague-Dawley rats (Hunan Slack Jingda Co., Ltd, Changsha, China) about 200–250 g were allowed to adapt to the environment for at least 1 week before the experiment. The case temperature is maintained at constant temperature (22 ± 2 °C) under standard light conditions with 12 h light-12 h dark cycle, lights on at 07:00 am, and humidity of 35–40%. The Animal Research and Education Committee of the Animal Experimental Center of Renmin Hospital of Wuhan University approved the experimental procedure (WDRM.20170910). The study was conducted following the ethical guidelines recommended by the International Association for Pain Research. Efforts were made to minimize animal suffering.

According to different experimental purposes, the rats we studied were divided into two parts. One part, mainly constructing the rat models, compared the differences among them. They were the control group (*n* = 6), epilepsy group (*n* = 6), migraine group (*n* = 6), and epilepsy–migraine comorbidity group (*n* = 6). The other part comprised mainly conducted drug intervention experiments. It included a comorbidity group (C) (*n* = 3), comorbidities + TAK-242 group (C + TAK-242) (*n* = 3), and comorbidities + muscimol group (C + Muscimol) (*n* = 3).

### 2.2. Surgical Procedures

For the experiments, the rat models were established according to Table 1. Epilepsy, migraine, and comorbidity rat model were established according to previous studies [20]. Animals were anesthetized with 2% pentobarbital (Sigma, 40 mg/kg, dissolved in PBS) and fixed on the stereotaxic instrument (Shenzhen Ruiwo De Life Technology Co., Ltd. Shenzhen, China.).

Cut the skin and muscles along the midline of the head with blunt dissection of the fascia. A hand drill (Shenzhen Ruiwo De Life Technology Co., Ltd.) was used to drill a bone window with a diameter of 1 mm, 1.5 mm from the right superior sagittal sinus, and 1.5 from the transverse sinus, so as not to destroy the dura at the right anterior frontal bone. A guide cannula was fixed on the hole with the help of dental cement. Then, we sutured the incision and applied erythromycin for 3 d after surgery.

### 2.3. Establish LiCl-Pilocarpine Rat Epilepsy Model

On the first day after surgery, all animals were intraperitoneally (i.p) injected with 1.25% lithium chloride solution (LiCl, 125 mg/kg, dissolved in PBS, Soledad Takara Bio). Eighteen hours later, atropine (1 mg/kg, dissolved in 0.9% normal saline, Sigma) was injected subcutaneously to reduce the peripheral cholinergic effect of pilocapine and relieve the pain of the rats. Half an hour later, the rats were given another intraperitoneal injection of 1% pilocarpine to induce epilepsy (25 mg/kg, dissolved in PBS, Cayman Chemical). The stages of seizure degree were classified by the Racine scale [21]. The level 4 and above of Racine scale were included [22]. Ten minutes after the epileptic seizure reached its maximum level (4 or above), 2% pentobarbital was injected intraperitoneally to cease seizures. The control group was injected with the same dose of PBS.

### 2.4. Establish Migraine Model with Inflammatory Soup

The third day after the procedure, 30 µL “inflammatory soup” (IS), mixed with 1 mM bradykinin (Soledad Takara Bio), 5-hydroxytryptamine (Soledad Takara Bio), histamine (Soledad Takara Bio), and 0.1 mM prostaglandin E2 (Sigma), dissolved in PBS (pH 5.5), was injected into the guide cannula to estimate the migraine models. The same dose of IS was injected daily 24 h before the interval. The control and epilepsy animals were injected with PBS instead.

### 2.5. Head-Scratching and Pain Threshold Recording

After the last application of IS, the number of head-scratching rats was recorded within the first 1 h. The pain threshold was measured at the skin behind the right eye corner of the rat with von Frey filaments. A positive manifestation on the von Frey test was recorded.

When measured 5 times and at least 3 times positive, the filament weight used was the pain threshold. The test starts with a mid-weight fiber filament. In order to protect the rats, those who did not respond to the weight of more than 15 g will record the pain threshold as 15 g. Each test was performed at least 10 s apart to allow the rats to recover from the previous stimulation response fully. The upper and lower limits of the fiber strength are 15 g and 0.4 g, respectively.

### 2.6. Drug Application

We used TAKC-242 as TLR4 inhibitor, which can selectively bind TLR4 and block the interaction between TLR4 and joint molecules, thus inhibiting TLR4 signal transduction and its downstream signal pathways [23]. Muscimol is a potent GABAA receptor agonist. Intramuscular injection of muscimol inhibits acetylcholine, histamine, or the vagus nerve by activating GABAA channels [24].

From the second day, before administration of pilocarpine or IS, TAK-242 (3 mg/Kg, dissolved in DMSO, APExBIO) and muscimol (1 mg/Kg, dissolved in sterile distilled water, R&D Systems) were injected intraperitoneally before 1.5 h and 0.5 h, respectively (Table 2). The epileptic latency and seizure levels of rats were recorded during the epilepsy period, and the number of scratches and the pain thresholds of rats was observed during the comorbid period.

### 2.7. Western Blotting

After the pain threshold examination was completed, half of the rats were transcardially perfused with 4 °C 0.1 M PBS (Phosphate buffered solution) 200 mL rapidly, and the temporal cortex, hippocampus, trigeminal ganglion, and medullary dorsal horn tissues were dissected out on the ice, and then stored at −80 °C. The tissues of rat brains were homogenized in RIPA lysis buffer (Beyotime Institute of Biotechnology, Shanghai, China) and centrifuged at 4 °C and 12,000 rpm for 10 min. Equal amounts of protein were resolved by SDS-PAGE electrophoresis, and wet-transferred to PVDF membranes. Membranes were blocked in 5% non-fat milk (10 mM Tris, pH 7.6, and 150 mM NaCl) for 1 h and probed overnight at 4 °C with primary antibodies. Rabbit-derived antibody dilutions of TLR4 (1:1000, Beyotime Institute of Biotechnology, Shanghai, China), GABAARα1 (1:1000, Abcam, Beyotime Institute of Biotechnology, Shanghai, China) or GAPDH (1:20,000, Beyotime Institute of Biotechnology, Shanghai, China) were used. Blots were then incubated with HRP goat anti-rabbit secondary antibody dilution (1:5000, Abbkine, Google Biotechnology Co., LTD, Wuhan, China). Immunoreactive proteins were detected with an enhanced chemiluminescence detection system. The relative density of the protein bands was quantified by densitometry and calculated for further statistical analysis.

### 2.8. Immunofluorescence Staining

After anesthesia, the rats were perfused with 0.1 m PBS (PH 7.4). After reperfusion with 4% paraformaldehyde, the rats were sacrificed and the brain tissues were stored at 4 °C for 24 h. The samples were processed through routine and standard paraffin embedding and were serially sectioned in 20 μm thickness, and then incubated with rat TLR4 (1:500, Abcam), rabbit GABARα1 (1: 500, Abcam), and mouse monoclonal OX42 (FITC-conjugated, 1:20, GeneTex, Google Biotechnology Co., LTD, Wuhan, China) antibodies. Then, goat anti-rat (1:1000, Abbkine) and goat anti-rabbit (1:1000, Thermo Fisher Scientific, Google Biotechnology Co., LTD, Wuhan, China) secondary antibodies were applied. Quantitative analyses of the fluorescence intensity were performed using ImageJ software (NIH). Measurements were recorded from randomly selected regions.

### 2.9. Co-Immunoprecipitation

We pre-incubated 5 µg of primary antibody (rat anti-TLR4, Abcam; rabbit anti-GABARα1, Abcam; rat or rabbit IgG, Bioss, Beijing, China) with 20 µL per sample in the suspended Protein A + G Agarose (20 µL; Santa Cruz Biotechnology, Shanghai, China) for 8–12 h at 4 °C and washed it 5 times. Samples were incubated at 4 °C for 30 min and centrifuged for 20 min at 16,000× *g*. The pellet was discarded. Protein (≥500 µg) from lysate was then combined with Protein A + G Agarose (Beyotime, Shanghai, China, P2055) coupled with primary antibody and incubated for 24 h while gently mixing at 4 °C. Next, the mixture was pre-washed with cold GLB+ buffer 3–5 times, and proteins were eluted with prepared 1 × loading buffer by boiling for 5 min and centrifuging at 16,000× *g* for 2 min to obtain the supernatant, which was then subjected to SDS-PAGE.

### 2.10. Statistical Analysis

SPSS22.0 (IBM Corporation, Armonk, NY, USA) software was used for statistical analysis. Measurement data were expressed as the mean ± standard deviation (x ± s). A one-way ANOVA was performed for group comparisons. Univariate analysis was used to evaluate differences amongst the groups. Groups were compared using the minimum significant difference method (LSD) to obtain uniform data. Non-normal distribution data was analyzed using Dunnett’s tests. *p*-values ≤ 0.05 indicated significant differences.

## 3. Results

### 3.1. Changes in Behavioral Responses

We used pilocarpine to induce an epilepsy phenotype on the second day when establishing the comorbidity rat model. There was no difference in the latency and Racine scale level of epilepsy between the comorbidity group and the epilepsy group (data not shown).

IS was used to induce migraine in rats by injecting into the guide cannula. At the fifth day, the head scratching times of each rat were recorded within 1 h after the IS administration. As shown in Figure 1A (Table 3), with the rats in the control and epilepsy groups, there was no obvious head scratching, while the frequency of rats injected with IS in the migraine and comorbidity groups had surged. However, the comorbidity group was further increased compared with the migraine group. The difference was statistically significant (*p* < 0.05) (Figure 1A). Data from the pain threshold test showed that the application of IS decreased the reaction of a rat to von Frey filaments in the migraine and comorbidity groups, especially in the comorbidity groups (Figure 1B, Table 3).

### 3.2. Pilocarpine and Inflammatory Soup Disrupt TLR4 and GABAARα1 Expression Levels in Brain Tissue

The role of TLR4 in immune function and inflammation has been well established; however, the role of targeted treatment or prevention of chronic neuropathic pain with TLR4 antagonists is still under investigation [17]. GABAAR is the primary inhibitory neurotransmitter receptor in the brain and the reduction in GABAAR expression level may be one of the critical mechanisms accounting for the neurobiology of drug-resistant epilepsy [25], with a strong association between the reduction of GABAARα1 subunit expression with generalized epilepsy [26]. We selected epilepsy- and migraine-sensitive areas, temporal cortex, hippocampus, trigeminal ganglion, and medullary dorsal horn as the research objects. The expression of TLR4 and GABAARα1 proteins in these regions was analyzed.

The study found the TLR4 abundance increased remarkably in the temporal cortex (Figure 2A,B), hippocampus (Figure 2D,E), trigeminal ganglion (Figure 2G,H), and medullary dorsal horn (Figure 2J,K) as measured by western blotting, and significant differences were observed between the control and other three groups (*p* < 0.05). The TLR4 abundance was the highest in the comorbidity group, compared with the other (*p* < 0.05) (Figure 2). The western blotting results showed that the expression of GABAARα1 abundance of the detected tissues decreased significantly in the migraine, epilepsy, and comorbidity groups compared with the control group, which was more pronounced in the comorbidity group (*p* < 0.05) (Figure 2A,C,D,F,G,I,J,L). There were no significant differences in the abundance of TLR4 and GABAARα1 between the migraine group and the epilepsy group (*p* > 0.05) (Figure 2).

In addition, rats of the comorbidity group displayed a decreased integrated density of GABAARα1, as indicated by immunostaining in the temporal cortex, medullary dorsal horn, and trigeminal ganglion (Figure 3). We also examined the expression of TLR4 in these brain sites of different groups. The integrated density of TLR4 of the comorbidity group was higher than the other three groups (Figure 4), which basically followed the tendency in western blotting (Figure 2). After quantification, we found that in the migraine and epilepsy groups, GABAARα1 in the medullary dorsal horn and the trigeminal ganglion was more highly expressed in the epilepsy group, and TLR4 in the migraine group was higher (Figure 3C,D and Figure 4C,D). This trend is more pronounced than in Western blots. Furthermore, we also performed a magnification analysis on the cells and found that GABAARα1 was expressed in both cytoplasms and nuclei, while TLR4 was expressed in some cytoplasms (Figure 3 and Figure 4 magnifying cells).

### 3.3. Epilepsy–Migraine Comorbidity Leads to an Increase of Microglia

Microglia are the immune cells of the brain, having high plasticity and possessing multiple functional phenotypes. Microglia are an indispensable part of the inflammatory processes in experimental models and human epilepsy. In epilepsy models, the activation of microglia and astrocytes activation can result from seizures alone, without cell loss [27,28,29]. Similarly, microglia play an essential role in the modulation of pain within the central nervous system [30]. Activation of microglia leads to the release of multiple mediators that are pronociceptive, such as interleukin 1, tumor necrosis factor, nitric oxide, interleukin 6, prostaglandins, adenosine triphosphate, and excitatory amino acids [31]. In comparison to the control group, the expressions of OX42-positive microglia in the trigeminal ganglions of epilepsy, migraine, and comorbidity groups were increased prominently, especially in the comorbidity group (Figure 5). All these data suggested that microglia promoted the occurrence of epilepsy–migraine comorbidity.

### 3.4. GABAARα1 Binds TLR4 and Regulates Its Expression

For preliminary observation of the involvement of TLR4 and GABAARα1 in epilepsy–migraine comorbidity, we determined the TLR4 and GABAARα1 expression patterns in different tissues of the brain. To examine whether GABAARα1 activates the TLR4 signal through its ability to bind TLR4, the protein extracted from the medullary dorsal horn tissue of epilepsy–migraine comorbid rats was studied for TLR4/GABAARα1 co-immunoprecipitation [32]. Immunoprecipitation with normal IgG served as the control. The data summarized in Figure 6 indicate that TLR4 was present in the reciprocal GABAARα1 precipitates but not in the normal IgG precipitates, proving that GABAARα1 binds TLR4. Since GABAARα1 has the same molecular weight as the IgG heavy chain, both are 55 KDa. When the TLR4 antibody and IgG were co-immunoprecipitated, a band corresponding to 55 KDa was visible (data not shown).

Having seen that TLR4 and GABAARα1 were bonded to each other, we wanted to know whether the two interact with each other and participate in the disease process. The proteins extracted from the temporal cortex, hippocampus, medullary dorsal horn, trigeminal ganglion of the comorbidity, comorbidity + TAK242, and the comorbidity + muscimol groups were immunoblotted with antibodies to TLR4 or GABAARα1 followed by GAPDH, used as a loading control. The results were quantitated and are expressed as mean GAPDH-adjusted densitometric units ± SEM. The levels of TLR4 were significant (*p* < 0.05) lower in the comorbidity + TAK-242 and comorbidity + muscimol groups than in the comorbidity group. This was accompanied by a significant (*p* < 0.05) increase in the levels of GABAARα1 in the comorbidity + muscimol group than in the comorbidity + TAK-242 and comorbidity groups (Figure 7, temporal cortex (A–C), hippocampus (D–F), medullary dorsal horn (G–I), and trigeminal ganglion (J–L)). The above demonstrated that changes in GABAARα1 led to changes in TLR4, whereas TLR4 did not affect GABAARα1. Collectively, the data indicated that the neuronal GABAARα1 bound TLR4, and regulated its expression, likely contributing to signal activation in epilepsy–migraine comorbidity.

### 3.5. Activates GABAARα1 or Interferes with TLR4 Activity, Affecting Behavior in Comorbid Rats

We subsequently used muscimol to activate GABAARα1 or TAK-242 to inhibit TLR4, respectively, to examine the effects of both on rat behavior. The use of the drug was shown in Table 2. The seizure latency and seizure levels of rats were recorded during the seizure period. Observation of the number of scratches and the facial mechanical withdrawal threshold were finished during the comorbid period. Statistical analysis showed that the GABAARα1 activation in the comorbidity + muscimol group significantly extended the seizure latency (Figure 8A), while the Racine scale level of epilepsy reduced (Figure 8B) (*p* < 0.05). Moreover, there was no significant difference (*p* > 0.05) between the comorbidity + TAK-242 group and the comorbidity group (Figure 8A,B, Table 4). The facial mechanical withdrawal threshold and the number of scratches of the comorbidity were also performed during the comorbid period. Compared with the comorbidity group, both TAK-242 and muscimol resulted in a decreased number of scratches and increased facial mechanical withdrawal thresholds (*p* < 0.05) (Figure 8C,D, Table 5). Combined data indicated that, in the epilepsy–migraine comorbidity group, GABAARα1 was involved in regulating seizures and migraine attacks, while TLR4 was involved in migraine attacks. The GABAARα1-TLR4 signal regulated the behavior of the comorbidity.

## 4. Discussion

Epilepsy and migraine are among the most prevalent neurological disorders. The two have several characteristics in common, including specific clinical features, overlapping pathophysiological mechanisms, and treatment methods. In some syndromes, they may also be associated with genetic and molecular substates. Comorbidity makes the existence of one disease increase the likelihood of the other. Therefore, the diagnosis and treatment of each disease and the potential presence of the other must be considered. Providers must be conscious of epilepsy–migraine comorbidity, its clinical spectrum, and its therapeutics. We reported that the vulnerability involved in epilepsy–migraine comorbidity was related to the GABAARα1 subunit in distinct brain sites, and a pathway regulated by GABAARα1 included TLR4 in different brain regions. Furthermore, epilepsy–migraine comorbidity was characterized by microglial activation. To the extent of our knowledge, this article is the first to highlight that the innate immune receptor TLR4 is controlled by GABAARα1 in the brain and is associated with a predisposition to epilepsy–migraine comorbidity.

To better evaluate the contribution of GABAARα1 and TLR4 expression patterns to epilepsy–migraine comorbidity, we used the LiCl-pilocarpine rat epilepsy model combined with inflammatory soup to establish the comorbidity rat model. The comorbidity group manifested a profound augment in head scratching, and a noticeable decrease in facial mechanical withdrawal threshold compared with the control group. Additionally, the migraine group also followed this trend, and it was not significantly different from the epilepsy group. Comorbid rats significantly differed from the other three groups, indicating that epilepsy–migraine comorbidity was not a simple superposition of the two diseases.

The inhibitory neurotransmitter GABA is widely distributed throughout the nervous system [33]. The decrease in neuronal GABA content causes an increase in neural network excitability, which, in turn, induces abnormal high-frequency discharge [34]. Among all the GABAAR subunits, the GABAARα1 is the most widely expressed in neurons of the brain with high expression levels [35]. The temporal cortex, hippocampus, trigeminal ganglion, and medullary dorsal horn were chosen as research objects for their sensitivity to epilepsy and migraine. Using western blotting and immunofluorescence, we found the expression of GABAARα1 in different brain sites, which was reduced profoundly in the epilepsy and migraine groups, especially in the comorbidity group. These results indicated a high-frequency discharge in the comorbidity group in these brain areas.

Epilepsy and migraine are diseases caused by abnormally increased neuronal excitability, and cortical spreading depression (CSD) is considered to be the connection between them [36,37]. CSD is an electrophysiological neuronal inhibition phenomenon caused by excessive depolarization of local cerebral cortical neurons. The depolarization phase is associated with an increase in cerebral blood flow, whereas the phase of reduced neural activity is associated with a reduction in flow. This is thought to cause activation of trigeminal nerves and subsequent release of neuroinflammation mediators [38]. It can be postulated that decreases in GABA content cause changes in neuronal excitability, induce CSD, and cause epilepsy, migraine, and comorbidity.

In these sensitivity brain regions, the expression of TLR4 showed an opposite trend to that of GABAARα1, which was the highest in the comorbid group. It proved that TLR4 was involved in the processes of epilepsy, migraines, and epilepsy–migraine comorbidity, especially the latter two. Moreover, using immunofluorescence, both TLR4 and GABAARα1 were expressed in the cytoplasm. It seemed reasonable to hypothesize the relationship between GABAARα1 and TLR4. The co-immunoprecipitation uncovered that GABAARα1 indeed binds TLR4. Moreover, activating GABAARα1 inhibited TLR4, and recovered the behavior of epilepsy and migraine. Inhibiting TLR4 did not affect GABAARα1 expression, and improved migraine only. The above indicated that the epilepsy–migraine comorbidity was a result of overinhibited GABAARα1 in the brain; the neuronal GABAARα1 bound and regulated TLR4; GABAARα1 was directly involved in the regulation of epilepsy, and a negative effect on the relative TLR4 participated in migraine.

TLR4 is involved in the innate immune inflammatory pathway of humans. In the nervous system, it is mainly expressed in microglia and astrocytes [39,40]. In this study, we found increased microglial activation in epilepsy–migraine comorbidity. The connection between GABAARα1 and TLR4 could have autocrine or paracrine effects that involve glial cells, especially microglia, and neurons, all of which express TLR4. However, the contribution of TLR4 to the activation or sustaining of migraine, and its microglial regulation and mechanisms that are involved in epilepsy–migraine comorbidity, is unknown. After TLR4 activation, NF-κB, IL-1β, and TNF-α are up-regulated, which promotes the activation of glial cells that produce inflammatory cytokines, resulting in a hyperalgesia state of the body [41,42]. Innate immune-induced neuroinflammation also increases the severity of seizures and increases the frequency of recurrence [43]. Our findings follow on from recent studies that suggest that (i) the GABAARα1-TLR4 axis contributes to epilepsy–migraine comorbidity, (ii) TLR4 is a critical intermediate link in epilepsy–migraine comorbidity, and (iii) immune-induced neuroinflammation in microglia may be involved in epilepsy–migraine comorbidity.

In this study, although the establishment of this comorbidity model has been proved to be effective and feasible, there are few examples of disease research using this model, and more effective comparisons cannot be made. An important question is, why does TLR4 contribute to epilepsy–migraine comorbidity downstream of GABAARα1? Is epilepsy–migraine comorbidity related to TLR4 microglial activation associated with CSD and inflammation? Future studies are needed to explore the role of TLR4 in CSD and to elucidate the potential role of inflammation-related chemokines/cytokines regulated by TLR4 in epilepsy–migraine comorbidities. The epilepsy–migraine comorbidity model, using inflammatory soup based on the LiCl-pilocarpine rat epilepsy model, provides a useful tool for a follow-up test to solve epilepsy–migraine comorbidity, and may find good treatment strategies to improve disease symptoms.

## Figures and Tables

**Figure 1 brainsci-12-01436-f001:**
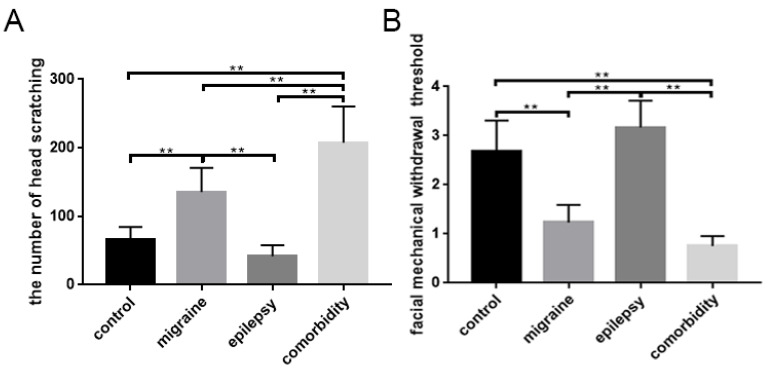
Inflammatory soup makes migraine symptoms worse. Significant differences in the amount of head scratching (**A**) and facial mechanical withdrawal threshold (**B**) were observed among four groups (*n* = 6; ** *p* < 0.01).

**Figure 2 brainsci-12-01436-f002:**
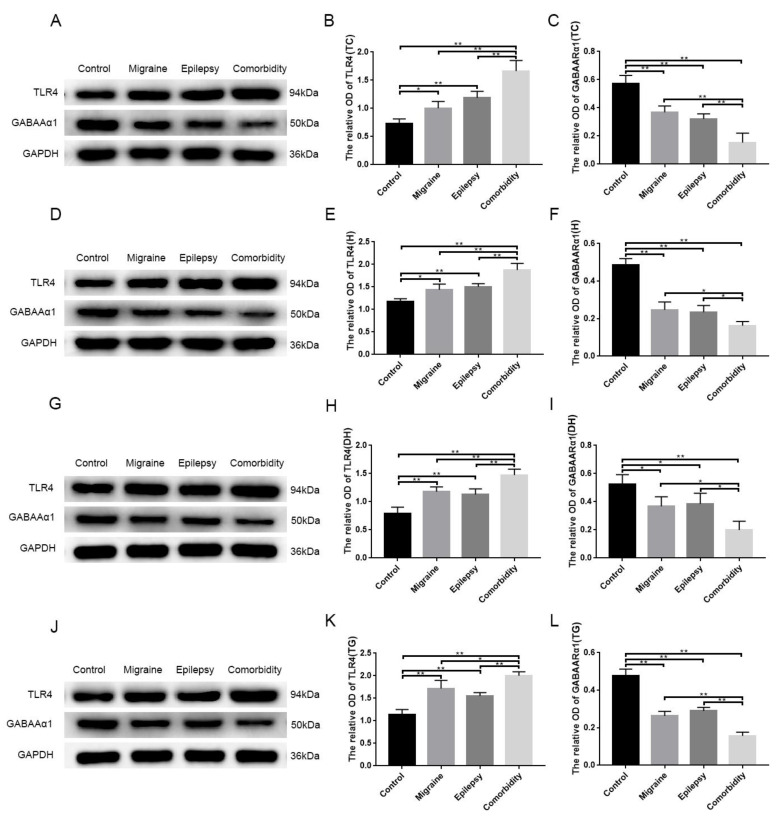
The disrupted expression of TLR4 and GABAARα1 in different brain sites. Protein extracts were immunoblotted with antibodies to TLR4, GABAARα1, and GAPDH. The blots were stripped between antibodies and results expressed as densitometric units ± SEM. Expression levels of TLR4 and GABAARα1 protein in the temporal cortex (**A**–**C**), hippocampus (**D**–**F**), medullary dorsal horn (**G**–**I**), and trigeminal ganglion (**J**–**L**) were determined using western blotting and quantitative assessment. Significant differences were observed between the epilepsy–migraine comorbidity group and the other three groups (*n* = 3, * *p* < 0.05, ** *p* < 0.01). There were no apparent differences between the migraine group and the epilepsy group. TC: temporal cortex, H: hippocampus, DH: medullary dorsal horn, TG: trigeminal ganglion.

**Figure 3 brainsci-12-01436-f003:**
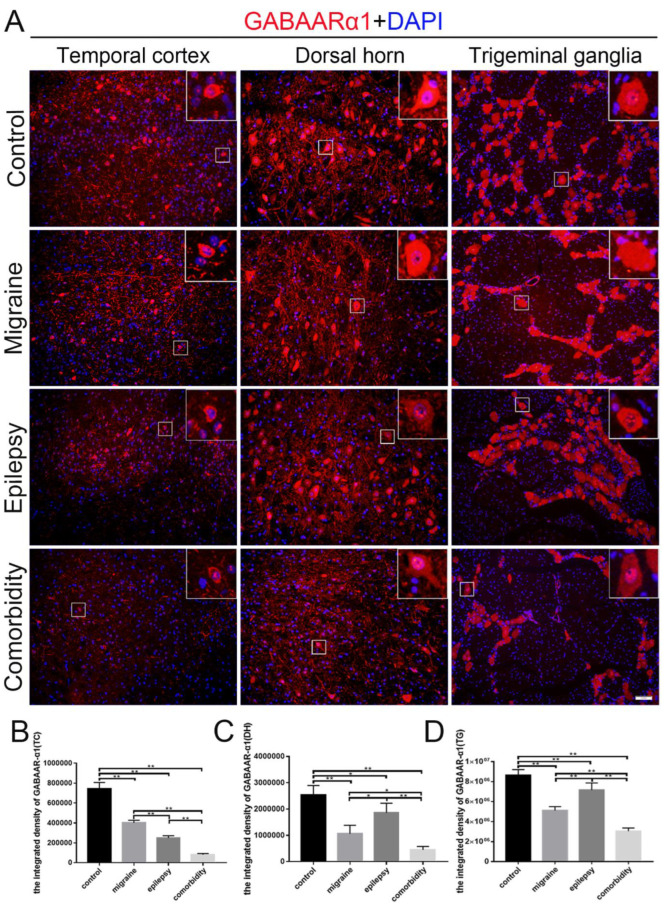
Localization of GABAARα1 in the temporal cortex, medullary dorsal horn, and trigeminal ganglion in four groups. (**A**) GABAARα1 (red) localized in the temporal cortex, medullary dorsal horn, and trigeminal ganglion in four groups. GABAARα1 was expressed in both the cytoplasm and the nucleus (**A**, magnifying cells). (**B**–**D**) Quantitative assessment of the integrated density of GABAARα1 in temporal cortex (**B**), medullary dorsal horn (**C**), and trigeminal ganglion (**D**) (*n* = 3, * *p* < 0.05, ** *p* < 0.01). DAPI (blue) indicates cell nuclei. The scale bar represents 50 μm. TC: temporal cortex, DH: medullary dorsal horn, TG: trigeminal ganglion.

**Figure 4 brainsci-12-01436-f004:**
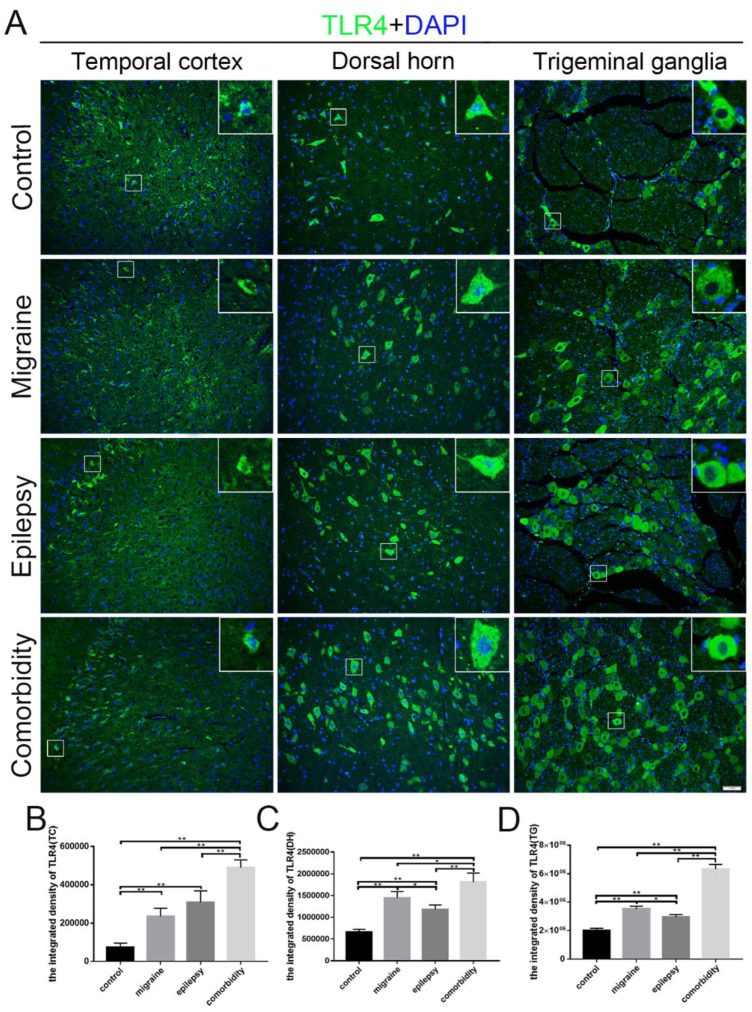
Localization of TLR4 in the temporal cortex, medullary dorsal horn, and trigeminal ganglion in four groups. (**A**) TLR4 (green) localized in the temporal cortex, medullary dorsal horn, and trigeminal ganglion in four groups. TLR4 was expressed in some cytoplasms ((**A**), magnifying cells). (**B**–**D**) Quantitative assessment of the integrated density of TLR4 in temporal cortex (**B**), medullary dorsal horn (**C**), and trigeminal ganglion (**D**) (*n* = 3, * *p* < 0.05, ** *p* < 0.01). DAPI (blue) indicates cell nuclei. The scale bar represents 50 μm. TC: temporal cortex, DH: medullary dorsal horn, TG: trigeminal ganglion.

**Figure 5 brainsci-12-01436-f005:**
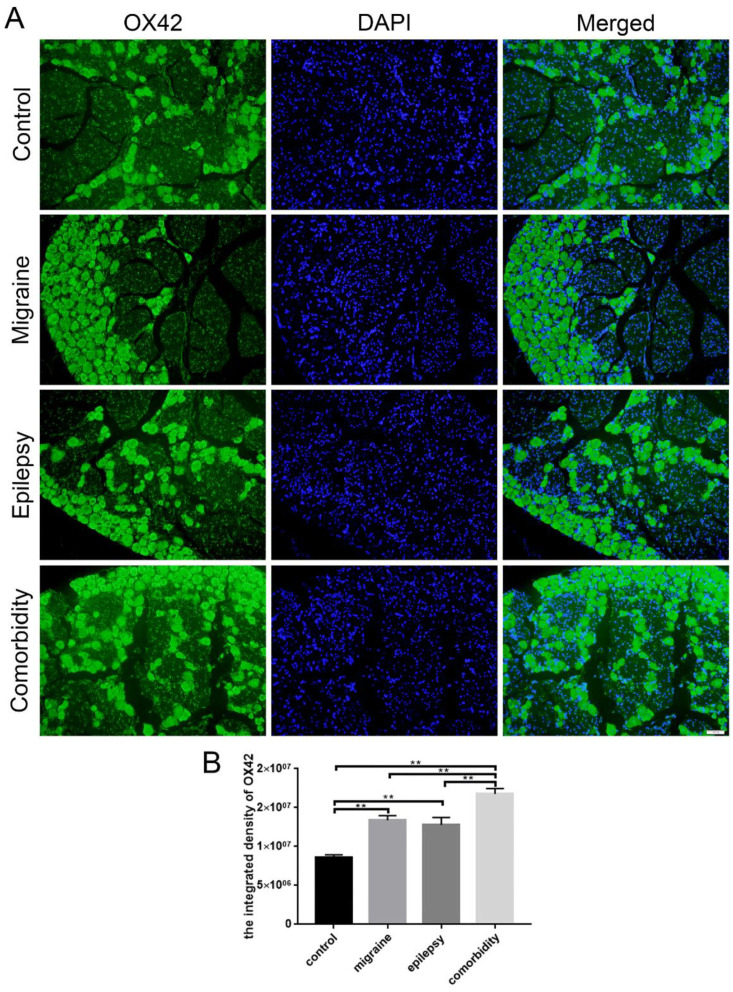
Localization of OX42 in the trigeminal ganglion in four groups. (**A**) OX42 (green) localized in the trigeminal ganglion in four groups. (**B**) Quantitative assessment of the integrated density of OX42 in the trigeminal ganglion (*n* = 3, ** *p* < 0.01). DAPI (blue) indicates cell nuclei. The scale bar represents 50 μm.

**Figure 6 brainsci-12-01436-f006:**
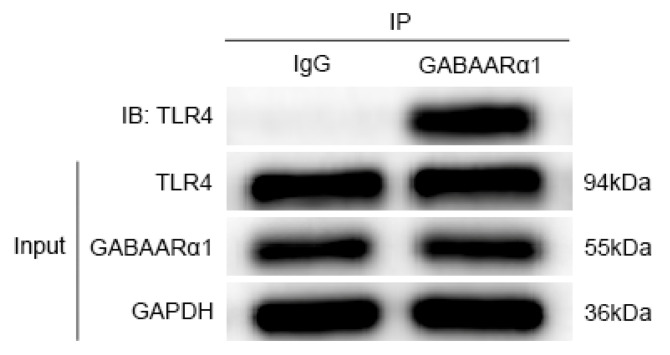
GABAARα1 binds TLR4 in the medullary dorsal horn of the epilepsy–migraine comorbidity. Protein extracts from the medullary dorsal horn tissue of the epilepsy–migraine comorbid rats were studied for TLR4/GABAARα1 co-immunoprecipitation (co-IP) with the GABAARα1 antibody or normal IgG (control), and the precipitates were immunoblotted with GABAARα1, TLR4, and GAPDH antibodies. TLR4 was seen in the anti-GABAARα1 (but not normal IgG) precipitates from the protein extracts, indicative of protein–protein interaction.

**Figure 7 brainsci-12-01436-f007:**
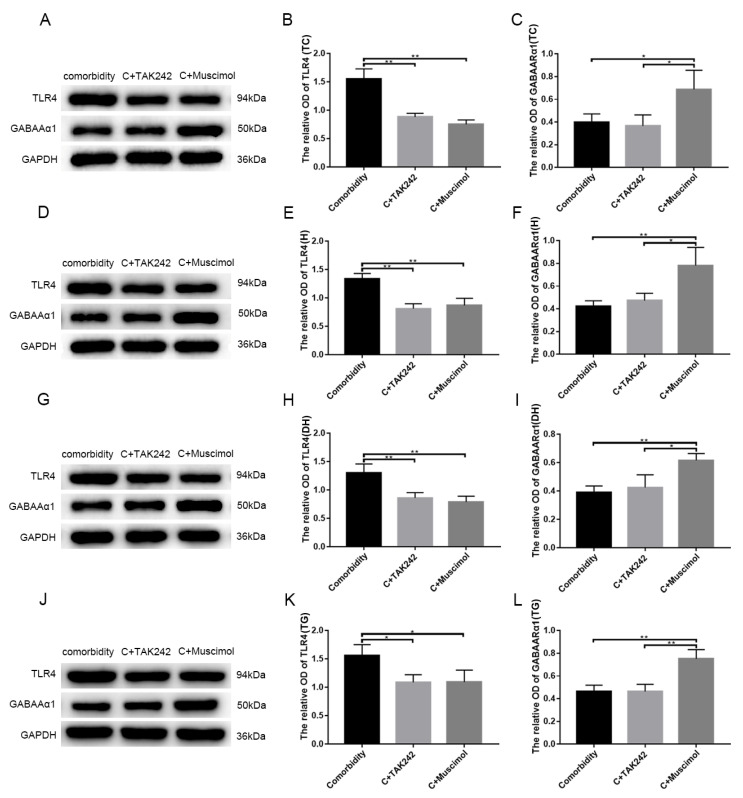
GABAARα1 negatively regulates TLR4 expression. Using TAK-242 to inhibit TLR4 or Muscimol to activate GABAARα1, Protein was collected from the temporal cortex (**A**), hippocampus (**D**), medullary dorsal horn (**G**), and trigeminal ganglion (**J**) of comorbidity, comorbidity + TAK-242, and comorbidity + Muscimol rats. Protein extracts were immunoblotted with antibodies to TLR4, GABAARα1, GAPDH used as a gel loading control. The results are expressed as densitometric units normalized to GAPDH ± SEM. The TLR4 levels were lower in comorbidity + TAK-242 and comorbidity + Muscimol groups compared with the comorbidity group (*n* = 3, * *p* < 0.05, ** *p* < 0.01) (**B**,**E**,**H**,**K**). The GABAARα1 levels were higher in comorbidity + Muscimol group compared with the comorbidity and comorbidity + TAK-242 groups (*n* = 3, * *p* < 0.05, ** *p* < 0.01) (**C**,**F**,**I**,**L**). TC: temporal cortex, H: hippocampus, DH: medullary dorsal horn, TG: trigeminal ganglion, C: comorbidity.

**Figure 8 brainsci-12-01436-f008:**
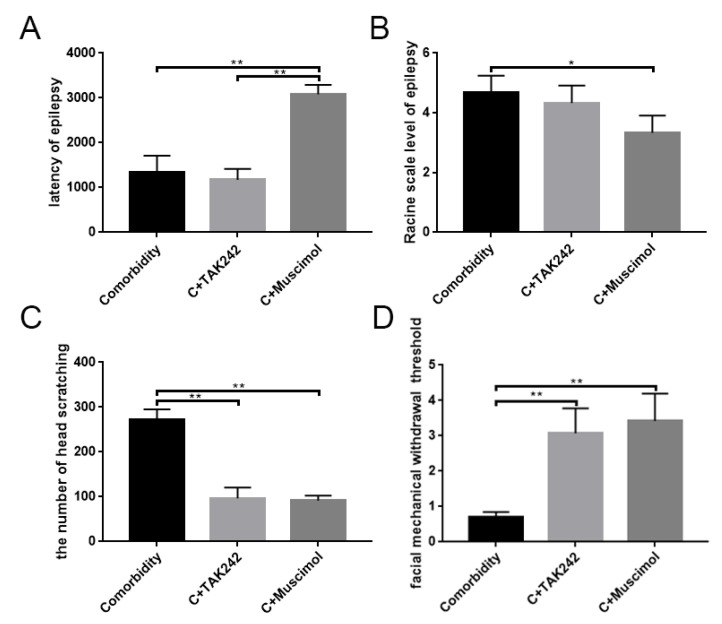
The GABAARα1-TLR4 signal regulates epilepsy–migraine comorbidity. (**A**) Significant differences in latency of epilepsy were observed between comorbidity (*n* = 3) and comorbidity + Muscimol (*n* = 3, ** *p* < 0.01), comorbidity + TAK-242 (*n* = 3) and comorbidity + Muscimol (*n* = 3, ** *p* < 0.01). No significant difference between comorbidity and comorbidity + TAK-242. (**B**) Application of Muscimol reduces the Racine scale level of epilepsy in comorbidity (*n* = 3, * *p* < 0.05). The amount of head scratching (**C**) and facial mechanical withdrawal threshold (**D**) were evaluated, and phenotype improvements were detected from comorbidity + TAK-242 (*n* = 3) and comorbidity + Muscimol rats (*n* = 3, ** *p* < 0.01). C: comorbidity.

**Table 1 brainsci-12-01436-t001:** Establishment of animal model.

	Day 1	Day 2	Day 3	Day 4	Day 5
Control group (*n* = 6)	Surgery + PBS (I.P) (2:00 pm)	Saline (S.C) (08:00 am) PBS (I.P) (08:30 am)	PBS (08:30 am)	PBS (08:30 am)	PBS (08:30 am)
Migraine group (*n* = 6)	Surgery PBS (I.P)	Saline (S.C) (08:00 am) PBS (I.P) (08:30 am)	IS (08:30 am)	IS (08:30 am)	IS (08:30 am)
Epilepsy group (*n* = 6)	Surgery LiCl (I.P)	Atropine (S.C) (08:00 am) PILO (I.P) (08:30 am)	PBS (08:30 am)	PBS (08:30 am)	PBS (08:30 am)
Comorbidity group (*n* = 6)	Surgery LiCl (I.P)	Atropine (S.C) (08:00 am) PILO (I.P) (08:30 am)	IS (08:30 am)	IS (08:30 am)	IS (08:30 am)

Abbreviation: I.P, intraperitoneal injection; S.C, subcutaneous injection; LiCl, lithium chloride; PILO, pilocarpine; IS, inflammatory soup.

**Table 2 brainsci-12-01436-t002:** Establishment of animal model (drug intervention).

	Day 1	Day 2	Day 3	Day 4	Day 5
Comorbidity group (n = 3)	Surgery + LiCl (I.P) (2:00 pm)	DMSO) (07:00 am) + Sterile distilled water (08:00 am) (I.P) + PILO (I.P) (08:30 am)	DMSO (07:00 am) + Sterile distilled water (08:00 am) (I.P) + IS (08:30 am)	DMSO (07:00 am) + Sterile distilled water (08:00 am) (I.P) + IS (08:30 am)	DMSO (07:00 am) + Sterile distilled water (08:00 am) (I.P) + IS (08:30 am)
Comorbidity +TAK-242 group (n = 3)	Surgery + LiCl (I.P) (2:00 pm)	TAK-242 (07:00 am) + Sterile distilled water (08:00 am) (I.P) + PILO (I.P) (08:30 am)	TAK-242 (07:00 am) + Sterile distilled water (08:00 am) (I.P) + IS (08:30 am)	TAK-242 (07:00 am) + Sterile distilled water (08:00 am) (I.P) + IS (08:30 am)	TAK-242 (07:00 am) + Sterile distilled water (08:00 am) (I.P) + IS (08:30 am)
Comorbidity + Muscimol group (n = 3)	Surgery + LiCl (I.P) (2:00 pm)	DMSO (07:00 am) + Muscimol (08:00 am) (I.P) + PILO (I.P) (08:30 am)	DMSO (07:00 am) + Muscimol (08:00 am) (I.P) + IS (08:30 am)	DMSO (07:00 am) + Muscimol (08:00 am) (I.P) + IS (08:30 am)	DMSO (07:00 am) + Muscimol (08:00 am) (I.P) + IS (08:30 am)

Abbreviation: I.P, intraperitoneal injection; LiCl, lithium chloride; PILO, pilocarpine; IS, inflammatory soup; DMSO, dimethylsulfoxide.

**Table 3 brainsci-12-01436-t003:** Comparison of the amount of head scratching and facial mechanical withdrawal threshold in different groups of rats.

	Control Group (*n* = 6)	Migraine Group (*n* = 6)	Epilepsy Group (*n* = 6)	Comorbidity Group (*n* = 6)
the amount of head scratching	66.33 ± 18.21 (times)	135.50 ± 35.35 (times)	41.83 ± 16.13 (times)	207.67 ± 52.78 (times)
facial mechanical withdrawal threshold (square root transformation)	2.68 ± 0.63 (√g)	1.23 ± 0.36 (√g)	3.16 ± 0.55 (√g)	0.75 ± 0.20 (√g)

**Table 4 brainsci-12-01436-t004:** The latency and Racine scale level of epilepsy in different groups of rats.

	Epilepsy Group (*n* = 3)	Epilepsy + TAK-242 Group (*n* = 3)	Epilepsy + Muscimol Group (*n* = 3)
The latency of epilepsy	1338.0 ± 372.04 (s)	1172.0 ± 244.83 (s)	3083.0 ± 208.62 (s)
Racine scale level of epilepsy	4.67 ± 0.58 (level)	4.33 ± 0.58 (level)	3.33 ± 0.58 (level)

**Table 5 brainsci-12-01436-t005:** Comparison of the number of head scratching and facial mechanical withdrawal threshold in different groups of rats after drug intervention.

	Comorbidity Group (*n* = 3)	Comorbidity + TAK-242 Group (*n* = 3)	Comorbidity + Muscimol Group (*n* = 3)
The amount of head scratching	271.67 ± 23.46 (times)	96.67 ± 23.71 (times)	91.33 ± 11.15 (times)
Facial mechanical withdrawal threshold (square root transformation)	0.71 ± 0.14 (√g)	3.07 ± 0.71 (√g)	3.42 ± 0.78 (√g)

## Data Availability

Not applicable.

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
