# Peer review of "Downregulation of GABAARα1 Aggravates Comorbidity of Epilepsy and Migraine via the TLR4 Signaling Pathway"

_brainsci, 2022, doi:10.3390/brainsci12111436_

Round 1

Reviewer 1 Report

Dear Editor, Dear Authors,

The way the manuscript is written makes it difficult to understand in several places.

Some examples of aspects that should be better specified:

a) some acronyms are not detailed;

b) a concise explanation of the rationale of the “inflammatory soup” model (hypersensitivity and activation of the trigeminal system?);

c) the distinction between headache and migraine is not always clear (as a lines 40-41: Epilepsy-migraine comorbidities are classified into epileptic headache, pre-seizure headache, post-seizure headache, and inter-epileptic headache);

d) what is meant by: "usually occurring before prodrome and/or aura" (lines 39-40); "The same dose of IS was injected daily 24 hours before the interval" (which interval?) (lines 116-117); what does "(time)" indicate in the 4 columns of table 3; line 385: “trigeminal ganglion, and medullary dorsal horn” are rich in GABA receptors, but does this involve their “sensitivity to epilepsy”?; line 31: “About 12% of people have migraines, and 2% of them have chronic migraines”: 2% of them or of the global population?; substates? (line 361).

Those shown are just examples of often imprecise writing. Consequently, I am unable to express any judgment on the validity and relevance of the experiments presented.

Sincerely.

Author Response

Dear Reviewer ,

Thank you for giving us the opportunity to submit a revised draft of the manuscript “Downregulation of GABAARα1 aggravates comorbidity of epilepsy and migraine via the TLR4 signaling pathway” for publication in the Journal of Brain Sciences. We appreciate the time and effort that you and the reviewers dedicated to providing feedback on our manuscript and are grateful for the insightful comments on and valuable improvements to our paper.

We have incorporated most of the suggestions made by the reviewers. Those changes are highlighted within the manuscript. Please see below, in blue, for a point-by-point response to the reviewers’comments and concerns. All page numbers refer to the revised manuscript file with tracked changes.

  1. We have read through this article again at your request. The abbreviations in this article have been modified. It is believed that the abbreviations in the revised version meet the requirements. Please check. If there is any problem, please feel free to inform us for modification.
  2. In the revised draft, we added the specific introduction of inflammatory soup, including its composition, mechanism of action and application in previous studies, please refer toline
  3. Here, the "headache" in the pre-seizure, post-seizure, etc., is actually a migraine, which is simply referred to as a headache. This is what the previous literature has written, so I wrote it as well. In the revised draft, we have made a new explanation, please check(line 41-43).
  4. The first "interval" means to give the drug at the same time every day, so it means to give it every 24 hours. The specific time point of drug administration has been marked in the table to facilitate a more intuitive understanding of the drug administration process.
  5. In Table 3, the meaning of "time" is the number of times the rat scratches its head.
  6. The rich GABA receptors in trigeminal ganglion and dorsal horn of medulla oblongata are only the results found in our experiment, and whether they are related to epilepsy sensitivity remains to be studied in the future.
  7. “2%”means 2% of the people who get migraines.

Thank you again for your positive comments and valuable suggestions to improve the quality of our manuscript. If there are any other modifications we could make, we would like very much to modify them and we really appreciate your help. Thank you very much for your help.

Your sincerely.

Reviewer 2 Report

The subject is interesting. The paper might be more significant if the authors had provided a more extensive discussion of the various techniques. However, certain aspects of the paper's technicality and presentation prevent me from accepting it, and I am marking it as the "Major Revision."

1- Conclusions: What is stated in the conclusion section is not fully supported by the data shown in the paper.

2-  Please explain more about the paper’s main contribution.  

3- The comparative study from the recently proposed method is missing. 

4- The author needs to explain how the method helps the current field and what challenges it solves. 

5- The literature review in the article seems insufficient. Similar approaches should be included in the introduction section or given as a separate section.

6The results and discussions section should be reorganized in a more highlighting, argumentative way. I strongly recommend adding a comparison with some recent studies.

Author Response

Dear Reviewer ,

Thank you for giving us the opportunity to submit a revised draft of the manuscript “Downregulation of GABAARα1 aggravates comorbidity of epilepsy and migraine via the TLR4 signaling pathway” for publication in the Journal of Brain Sciences. We appreciate the time and effort that you and the reviewers dedicated to providing feedback on our manuscript and are grateful for the insightful comments on and valuable improvements to our paper.

We have incorporated most of the suggestions made by the reviewers. Those changes are highlighted within the manuscript. Please see below, in blue, for a point-by-point response to the reviewers’comments and concerns. All page numbers refer to the revised manuscript file with tracked changes.

  1. In this study, through animal modeling(In fact, animal modeling is to verify the feasibility of the comorbidity model, and our group has confirmed the feasibility of this modeling method in the previous study,cite Fan S, Xiao Z, Zhu F, He X, Lu Z. A new comorbidity model and the common pathological mechanisms of migraine and epilepsy. Am J Transl Res. 2017;9(5):2286–2295), WB, immunofluorescence, co-immunoprecipitation and other experimental methods, it was found that: i) GABAARα1-TLR4 axis was related to epileptic-migraine comorbidities; (ii) TLR4 is a key intermediate link in epileptic-migraine comorbidities; (iii) Neuroinflammation induced by microglial immunity may be associated with epileptic-migraine comorbidity. 
  2. In my opinion, the role of TLR4 and GABA-a1 pathway in the comorbidity of epilepsy and migraine has been discovered, which provides more ideas for the study of comorbidity.
  3. In this study, we found that epileptic-migraine comorbidity is related to GABAARα1-TLR4, and TLR4 is a key link. In other words, the comorbid rat model is used to provide a new research idea. As for solving the problem, I think this is only a preliminary study, which has not risen to the level of solving the problem.
  4. In the Introduction part of the manuscript, we added appropriate content, including the introduction and research of IS, to make the argument more sufficient.
  5. The comorbidity mouse model involved in this study was found to be rarely studied after reviewing relevant literature, only the previous research articles published by our research group.

Thank you again for your positive comments and valuable suggestions to improve the quality of our manuscript. If there are any other modifications we could make, we would like very much to modify them and we really appreciate your help. Thank you very much for your help.

Your sincerely.

Reviewer 3 Report

I have reviewed this manuscript the overall contents of this manuscript is well organized to give a clear overview of this work. I have suggested some comments about this work are as the following:

Comments to the Authors:

11.     Authors should write clearly abstract clearly including, background, method, results, significance of the study.

22.     In method section author should add one more figure related to the system architecture diagram/flowchart.

33.      My suggestion is that the authors should write discussion section clearly in more details like how and why this study is important than previous clinical research about Epilepsy, Migraine and comorbidity, like “Chikara RK, Chang EC, Lu Y-C, Lin D-S, Lin C-T and Ko L-W (2018) Monetary Reward and Punishment to Response Inhibition Modulate Activation and Synchronization Within the Inhibitory Brain Network. Front. Hum. Neurosci. 12:27. doi: 10.3389/fnhum.2018.00027”

44.     The authors should write some limitations of this study and clinical application in more details.

Author Response

Dear Reviewer ,

Thank you for giving us the opportunity to submit a revised draft of the manuscript “Downregulation of GABAARα1 aggravates comorbidity of epilepsy and migraine via the TLR4 signaling pathway” for publication in the Journal of Brain Sciences. We appreciate the time and effort that you and the reviewers dedicated to providing feedback on our manuscript and are grateful for the insightful comments on and valuable improvements to our paper.

We have incorporated most of the suggestions made by the reviewers. Those changes are highlighted within the manuscript. Please see below, in blue, for a point-by-point response to the reviewers’comments and concerns. All page numbers refer to the revised manuscript file with tracked changes.

  1. In the abstract of the revised draft, we reorganized the language to clearly express the methods, results, conclusions, significance and other parts of this study. Please refer to it.
  2. In my opinion, our study is based on previous studies, and by referring to previous studies, including modeling methods and research ideas, we have discovered the role of TLR4 and GABA-a1 pathways in the comorbidity of epilepsy and migraine, which can provide a new idea and method for future research. The discussion does not suggest that our study is any more important than previous studies.
  3. The specific time point of drug administration has been marked in the table to facilitate a more intuitive understanding of the drug administration process.Please refer to Table 1-2.
  4. In my opinion, our study is based on previous studies, and by referring to previous studies, including modeling methods and research ideas, we have discovered the role of TLR4 and GABA-a1 pathways in the comorbidity of epilepsy and migraine, which can provide a new idea and method for future research. The discussion does not suggest that our study is any more important than previous studies.
  5. I think the limitation is that it is only found in animal models, and it has not been demonstrated in human subjects. Moreover, the data are relatively small and coarse, and it is only a preliminary conclusion. If it is of clinical significance, I think some new research ideas can be found through the study of animal models. If some new action pathways, key factors and receptors can be found through the study, it may have certain inspiration for the development of new drugs. Please refer to line461-464.

Thank you again for your positive comments and valuable suggestions to improve the quality of our manuscript. If there are any other modifications we could make, we would like very much to modify them and we really appreciate your help. Thank you very much for your help.

Your sincerely.

Round 2

Reviewer 1 Report

Dear Authors,

some corrections have been made to make the manuscript more understandable. But some items, present in the 'Author response', have not been included. Some printing errors remain; examples: the acronyms TLR4 and GABAARα1 (line 16) are detailed only in lines 26 and 27; TAKC-242 (line 151).

Sincerely.

Reviewer 2 Report

The manuscript was modified very well. The authors have attempted to address all of the reviewers' comments in the revised paper. The manuscript seems acceptable to me for publication in the journal with the corrections made.